# NESTLE: An Efficient and Robust Data Valuation Framework for Large Language Models

## Abstract

The training and fine-tuning of large language models (LLMs) heavily rely on a large corpus of high-quality data. Nevertheless, the internet's extensive data is often of varying quality, and collecting high-quality data is exceedingly expensive. To facilitate data engineering and trading, the quantification of data value, also known as data valuation, is emerging as a critical topic. Traditional approaches for data valuation typically depend on model retraining. However, with the increasing model sizes and expansive data volumes of LLMs, these conventional methods are encountering significant declines in valuation precision, efficiency, and transferability. To alleviate these problems, we propose NESTLE, which is an efficient and robust framework for data valuation of LLMs. To accurately estimate the data value distribution across different target domains, we develop a training-free mechanism based on gradient tracing to simulate the data influences. To further tackle the dynamical value adjustment when multiple data providers coexist, we draw inspiration from the Shapley value theory and devise an accelerated strategy for estimating marginal contributions of data through gradient additivity. Extensive experiments demonstrate that our proposed framework NESTLE is capable of accurately and robustly providing accurate estimates of data value with a minuscule cost across a wide range of real-world scenarios.

## 1 Introduction

With the ongoing evolution of Large Language Models (LLMs) Touvron et al. (2023); Achiam et al. (2023); Zeng et al. (2023); Bai et al. (2023), it has become a common paradigm to finetune these LLMs with domain-specific data to align and enhance their downstream performances Zhou et al. (2023); Touvron et al. (2023), which largely hinges on large-scale, high-quality training data. However, the quality of publicly available data on the internet varies significantly, and the collection and curation of standard-compliant data are extremely time-consuming and labor-intensive Ghorbani & Zou (2019). To this end, data valuation Schoch et al. (2022); Ghorbani et al. (2020), which aims to estimate the worth of data from different sources, has gained considerable focus from the community and has been broadly utilized in real-world applications like model adaptation Jiang et al. (2023a) and data trading Agarwal et al. (2019); Jiang et al. (2023a).

Traditional data valuation approaches can be roughly grouped into marginal-contribution-based Ghorbani & Zou (2019); Jia et al. (2019) and influence-based Park et al. (2023); Jiang et al. (2023a); Pruthi et al. (2020). The former marginal-contribution-based ones typically assess the worth of data by quantifying the data contributions for different marginal subsets Schoch et al. (2022). The Leave-One-Out (LOO) Jia et al. (2019) strategy is achieved by tracking performance variations once different data sources are removed. Among these methods, the most prevalent paradigm is driven by the Shapley Value (SV) theory Shapley (1953); Schoch et al. (2022), which suggests arranging all possible subsets to assess the contribution of a specific data source (i.e., the SV for provider $i$ can be expressed as $\mathbb{E}_{S \subseteq N \setminus i}[U(S \cup i) - U(S)]$ where $S$ represents the collection of all possible subsets). Despite the success, these approaches can be computationally intensive, especially for large model sizes and expansive data volumes in the era of LLMs. These methods involve an atomic procedure of cumbersome retraining and testing for every data sources, and the SV manners even require $(2^N - 1)$ groups of such complicated atomic experiments for $N$ data providers.

On the other hand, another line of research relies on estimating the data influence Jiang et al. (2023a); Park et al. (2023) to alleviate the valuation cost for the marginal-contribution-based ones. These methods expedite the estimation of Shapley values by tracing gradient calculationsJiang et al. (2023a) or by assessing client contributions Tastan et al. (2024) during federated training processes. Nevertheless, these approaches either necessitate high-cost federated training across various data providers Koh & Liang (2017) or yield inaccurate estimates Jiang et al. (2023b), particularly in the presence of coexisting multiple data sources. Hence, such influnce-based approaches are not readily transferable to the data valuation of LLMs in multi-domain, multi-data source scenarios. It remains a challenge how to systematically estimate the value of different data sources for LLMs.

Motivated by this, we propose aN Efficient and robuSt daTa vaLuation framEwork, **NESTLE**, for LLMs in this work. Our primary target is to accurately and efficiently estimate the data value distribution of different data providers across different target domains. To achieve this, we first draw inspiration from traditional influence-based methods and then develop a training-free valuation mechanism that simulates the data influences in different domains based on gradient tracing. Specifically, we first maintain a customized support set for each of the target domains, which functions as the standard for data value estimation. Our core intuition is that the data value can be reflected by the decrease in the loss on the support set after the LLM is trained on the to-be-valuated data. We theoretically find that such a data influence can be further represented and formalized as the inner product of the gradient of the valuated data and that of the support set. Based on this theoretical foundation, we design our fundamental valuation framework, which consists of two steps: support data gradient caching and valuation data gradient querying. We further incorporate optimization strategies of gradient projection and gradient calibration to reduce the valuation cost and bolster its theoretical credibility. Empirically, such a framework can successfully and accurately estimate the value distribution of a single data source across different target domains.

Despite the success, such a framework fails to dynamically adjust the value distribution when multiple data providers coexist, for example, two data providers with highly similar data need to be penalized. To address this, we aim to harness the traditional Shapley value theory's intuition of marginal contribution to assess the interplay of multiple data providers by evaluating the marginal contributions of different subsets. Nevertheless, traditional Shapley value methods adopt performance metrics as the utility function and necessitate training for each marginal subset, resulting in exponential time complexity as the number of data providers increases. In contrast, our approach naturally evaluates value based on gradient sums rather than metrics. The additive nature of gradients compared to performance metrics can significantly reduce computational costs. By leveraging the additivity of gradients as a utility function in combination with the Shapley value, we can maintain the original linear time complexity without incurring exponential time costs. Extensive experiments demonstrate that our accelerated NESTLE framework can accurately adjust the data value distribution when multiple data providers coexist, while the time consumption is only 1.25% of the traditional ground-truth Shapley value. The contributions of this work are as follows.

- We delineate the task formulation, core requirements, and challenges of data value estimation for LLMs in multi-domain, multi-source scenarios. Then we proposed a training-free framework NESTLE based on gradient tracking to address these issues.
- Our framework estimates data value by tracking gradient inner product of support and valuation data. We further designed an accelerated Shapley-based valuation strategy, leveraging the additivity of gradients to handle dynamic value adjustments in multi-source scenarios.
- We conduct extensive experiments and demonstrate that our proposed framework can efficiently and accurately perform data value estimation across different task scenarios.

## 2 RELATED WORK

**Traditional Data Valuation.**  Measuring the value of training data is a pivotal theme in machine learning Sim et al. (2022); Ghorbani et al. (2020); Schoch et al. (2022), which focuses on assessing the worth of different data sources. It is broadly utilized in the applications of data trading Agarwal et al. (2019) and pricing Jiang et al. (2023a). Existing data valuation methods in machine learning can be mainly categorized into two types, marginal-contribution-based Ghorbani & Zou (2019); Jia et al. (2019) and influence-based Park et al. (2023); Jiang et al. (2023a); Pruthi et al. (2020). The former marginal-contribution-based approaches estimate the data worth by computing

its marginal performance improvement when the to-be-valued data is incorporated across different subsets Schoch et al. (2022). This can be achieved through computational paradigms such as the leave-one-out (LOO) Jia et al. (2019) and Shapley value Shapley (1953); Schoch et al. (2022). The latter method relies on data influence Koh & Liang (2017); Jiang et al. (2023a); Park et al. (2023) and aims to assess the impact of adding or removing specific data segments on model training Koh & Liang (2017). However, these approaches can lead to exceedingly unaffordable costs of training.

**Federated Client Contribution Estimation** Another relevant setting is the estimation of client contributions in federated learning Tastan et al. (2024); Jiang et al. (2023b); Li et al. (2020), which focuses on designing principled mechanisms to assess the contributions of individual participants from the standpoint of machine learning fairness Oneto & Chiappa (2019); Ghani et al. (2023). These methods predominantly build on Shapley value Koh & Liang (2017) and aim to develop more efficient estimation mechanisms Wang et al. (2020) of that.

**Data Selection for Large Language Models** The emergence of large language models (LLMs) has prompted initial research into exploring the significance of data quality during model fine-tuning Zhou et al. (2023); Li et al. (2024c); Chen et al. (2024); Xia et al. (2024). To enhance the quality of training data, the topic of data selection is receiving increasing attention Li et al. (2024b); Du et al. (2023); Ge et al. (2024). Some preliminary approaches are designed to filter high-quality training data through different perspectives such as difficulty Li et al. (2024b), quality Li et al. (2024a), diversity Ge et al. (2024), and necessity Du et al. (2023). The enhanced quality of the selected data then facilitates model training with improved performance and reduced training cost Xia et al. (2024). Some work additionally incorporates a validation set from downstream tasks as criteria to select training data that meet personalized requirements Li et al. (2024d); Xia et al. (2024). However, these data selection methods are primarily tailored to enhance the training efficiency, but fail to finely assess the value distribution of different data sources in various target areas. Thus, it remains a challenge how to systematically estimate the value of different data sources for LLMs.

## 3 METHOD

### 3.1 PRELIMINARIES

We primarily focus on data valuation for LLM fine-tuning across different targeted domains. Suppose that there is a set of $n$ data providers denoted as $N = \{1, \ldots, n\}$. The core objective of our data valuation framework is to estimate the value $\{\phi_k\}_{k \in N}$ of to-be-valuated dataset $D_{valu}^k$ for data provider $k$ on LLM $M_\theta$ across different targeted downstream domains $T = \{t_1, t_2, \ldots, t_m\}$. The estimated data value of $D_{valu}^k$ in the domain $t_j$ is then denoted as $\phi_k^j$. In targeted valuation settings, every target domain $t_j$ is equipped with a validation set (support set) $D_{sup}^j$, serving as a criterion for targeted data valuation. In summary, this framework aims to leverage validation sets $D_{sup}$ from different target domains $T$ to assess the value distribution of different data providers $N$ for the LLM $M_\theta$ across those domains.

To construct a comprehensive data valuation framework in such a scenario, we first discuss the requirements for an LLM data valuation framework. We believe that a comprehensive data valuation framework should meet the following design principles.

- **Accuracy**: The designed valuation framework needs to accurately estimate the value distribution of data across different domains. When multiple data providers coexist, precise dynamic adjustments are necessary to closely align with the ground-truth Shapley value.

- **Efficiency**: The valuation framework needs to measure data value effectively, preferably without costly training, while efficiently accommodating changes in data providers.

- **Adaptability**: The valuation framework needs to be broadly applicable, offering strong generality and flexibility for different domains, LLM architectures, and model sizes.

- **Robustness**: Based on several principles from existing works Agarwal et al. (2018); Ohrimenko et al. (2019); Bax (2019), we discuss the following necessary robustness requirements: *i) Strict Monotonicity.* if the dataset $D_k$ results in a more significant performance enhancement compared to $D_j$, then the value score of data owner $k$ should be strictly

higher than that of owner $j$; **ii) Symmetry.** If the dataset $D_k$ yields the same performance improvement as $D_j$ (e.g., $D_j = D_k$ ), then the value scores of these two datasets should be equal; **iii) Uselessness.** If the dataset $D_k$ fails to contribute to any performance enhancement, then $D_k$ should be valueless; **iv) Clone robustness.** If an provider $k$ participates in the collaboration with its duplicate $k^{'}$ (i.e. $D_{k'} = D_k$), the value allocated to $k$ (and $k^{'}$) should not increase; **v) Relevance.** If $D_j$ is similar to data from other sources and $D_k$ is unique to owner $k$, it is possible for $\phi_k$ to be greater than $\phi_j$ even if $U(D_k) \leq U(D_j)$.

## 3.2 Data Valuation via Gradient Tracing

Consider the LLM $M_\theta$ at time step $i$ with parameter $\theta_i$, We assume a parameter update process through meta-update, using a batch of samples $s_i \subset D_{valu}$ for a single training step. For this meta-iteration $a$, parameters of $M$ are updated from $\theta_i$ to $\theta_{i+1}$. Therefore, we define the to-be-estimated value of the mini-batch sample $s_o$, in relation to the support set $d_t \subset D_{sup}^t$ as,

$$V(s_i, d_t) = \mathcal{L}(d_t, \theta_a) - \mathcal{L}(d_t, \theta_{a+1}) \tag{1}$$

We can write the first-order Taylor expansion of this formula as,

$$\mathcal{L}(d_t, \theta_{a+1}) = \mathcal{L}(d_t, \theta_a) + (\theta_{a+1} - \theta_a)\nabla_{\theta_a}\mathcal{L}(d_t, \theta_a) \tag{2}$$

For ease of exposition, assume that we are training the model with SGD with batch size 1 and the step size of $\eta_a$

$$\theta_{a+1} = \theta_a - \eta_a \nabla_{\theta_a}\mathcal{L}(s_i, \theta_a) \tag{3}$$

Combining the above two equations Eq.( 2) and Eq.( 3), the influence of $s_i$ becomes,

$$V(s_i, d_t) = \eta_a \nabla_{\theta_a}\mathcal{L}(s_i, \theta_a)^\top \nabla_{\theta_a}\mathcal{L}(d_t, \theta_a) \tag{4}$$

For a particular training mini-batch $s_i$, we can approximate the influence by summing up this formula in all the iterations in which $s_i$ was used to update the parameters. Consequently, the impact of the data $D_{valu}$ on the support set $D_{sup}^t$ from domain $t$ can be assessed by summing their effects.

$$V(D_{valu}, D_{sup}^t) = \sum_{s_i \subset D_{valu}, d_t \subset D_{sup}^t} \eta_a \nabla_{\theta_a}\mathcal{L}(s_i, \theta_a)^\top \nabla_{\theta_a}\mathcal{L}(d_t, \theta_a)$$
$$= \bar{\eta_a}\nabla_{\theta_a}\mathcal{L}(D_{valu}, \theta_a)^\top \nabla_{\theta_a}\mathcal{L}(D_{sup}^t, \theta_a) \tag{5}$$

Consequently, the value of $D_{valu}$ within the target domain $t$ is estimated by the accumulation of the inner products of its gradient with those of the corresponding support set $D_{sup}^t$. In our valuation strategy, based on the idea of such gradient tracing, we develop an extended framework for value distribution estimation and dynamic adjustment, as outlined below.

## 3.3 Valuation Framework

In this section, we introduce our valuation framework, NESTLE, in detail. We primarily consider two key valuation scenarios. The first scenario involves a single data provider, where we propose a valuation strategy to quickly estimate the static value distribution of data in specific domains. The second scenario considers multiple data providers, where we need to account for interactions among them, thus proposing a dynamic adjustment mechanism based on Shapley-based approximation.

**Static Estimation of Value Distribution.** For static valuation, we primarily focus on the static estimation of value distribution for a single data provider. Relying on the gradient tracking theory from Eq.(5), we can estimate the value distribution of $D_{valu}$ across multiple domains $T = \{t_1, t_2, ..., t_m\}$ by computing gradient inner products between the valued data $D_{valu}$ and support sets $\{D_{sup}^{t_1}, D_{sup}^{t_2}, ..., D_{sup}^{t_m}\}$ for different target domains.

While this approach is straightforward and theoretically supported, there are still areas that need refinement and optimization. (i)-Firstly, this approach relies on gradient caching, requiring gradients to be stored at the sample, batch, or dataset level, with a cached tensor of shape $[len, n\_param]$, where $len$ is the number of gradient points and $n\_param$ is the number of trainable parameters.

Caching such large gradients for different valuation and support datasets is unaffordable for large LLMs, where the number of trainable parameters often reaches billions. To reduce the memory cost of gradient caching, we take inspiration from existing work Park et al. (2023) and incorporate an additional gradient projector , which reduces the gradient dimensions from $n\_param$ to 4096, thereby decreasing the memory overhead of gradient caching to about 0.01% of the original cost. (ii)-Second, the foundational gradient tracking theory rests on the SGD assumption outlined in Eq. (3). In practice, though, LLM training frequently uses Adam-like optimizers, necessitating a revision of assumption in Eq. (3). We apply an additional calibration mechanism, retaining the first and second moments of the gradients for Adam-style adjustments.

$$
\theta_{a+1} - \theta_a = -\eta_a \Gamma(s_a, \theta_a), \text{ where } \Gamma(s_a, \theta_a) = \frac{\mathbf{m}^{a+1}}{\sqrt{\mathbf{v}^{a+1}} + \epsilon}
$$
$$
\mathbf{m}^{a+1} = (\beta_1 \mathbf{m}^a + (1 - \beta_1)\nabla\ell(s_a, \theta_a))/(1 - \beta_1^a)
$$
$$
\mathbf{v}^{a+1} = (\beta_2 \mathbf{v}^a + (1 - \beta_2)(\nabla\ell(s_a, \theta_a))^2)/(1 - \beta_2^a)
$$
(6)

Through gradient projection and calibration, the lower-cost caching and more coherent theoretical assumptions help us achieve a higher-quality estimation of data value distribution.

**Dynamic Adjustment of Multi-Source Provider.** In the previous section, we propose a gradient tracing mechanism to estimate the data value distribution for a single data provider across different domains. However, in real-world data trading scenarios, multiple data providers often coexist, and their potential interactions might require dynamic adjustments to the value estimations. For instance, when the data from two providers partially or completely overlaps, we need to apply a reduction penalty to their data value. To achieve such dynamic correction, we draw inspiration from traditional Shapley value calculations and propose a dynamic value correction strategy based on rapid Shapley estimation. In particular, traditional Shapley value involves computing the additional contribution of target data $s_i$ to various marginal subsets $S$, where $S$ is a coalition that contains a subset from provided data source $N = \{D_{valu}^1, D_{valu}^2, ..., D_{valu}^n\}$. For data $D_{valu}^k$ from each provider $k$, the corresponding marginal set $S$ can be sampled from any subset from $N$ that does not contain $D_{valu}^k$, that is , $S \subseteq N \backslash \{D_{valu}^k\}$. The Shapley value of $D_{valu}^k$ is then formulated,

$$
v(D_{valu}^k) = \mathbb{E}_{S \subseteq N \backslash \{D_{valu}^k\}}[U(S \cup D_{valu}^k) - U(S)]
$$
(7)

where $U$ is the utility function that measures the performance of subset $S$ and $U(S)$ represents the data value of the subset $S$ (Traditional Shapley value typically utilizes performance metrics as utility function). Such computations of Shapley value are cost-intensive, as the marginal contribution needs to be calculated for all possible subsets, the required number of training iterations is 7 when there are 3 clients. In our gradient tracking framework, because model parameters aren't updated and we merely compute gradients for different valuation data batches, we can optimize computational complexity using the additivity property of gradients,

$$
grad(D_a \& D_b) = grad(D_a) + grad(D_b) - grad(D_a \cap D_b)
$$
(8)

By adopting this method, we can reuse computed gradients $grad(D_a)$ and $grad(D_b)$ from individual clients to calculate their union's gradient, simply subtracting the overlap $grad(D_a \cap D_b)$ . This dramatically cuts the cost of Shapley computations from the original $(2^n - 1)$ possible subsets to only $n$ data providers, allowing for dynamic adjustments at virtually no extra cost.

**Efficiency Analysis** The SV method, which relies on performance indicators, requires calculating the performance of all possible combinations of datasets from $n$ data owners. This involves training a model on each combination and subsequently evaluating its performance on a validation set. Hence, the time complexity escalates exponentially with the addition of more data owners (i.e., $O(2^n)$). In comparison, our NESTLE first calculates the gradients of different providers, and the gradients between data points are independent. When computing combinations of datasets $D_k$ and $D_j$, we can treat it as extending dataset $D_k$ by directly summing the representations of datasets $D_j$ and $D_k$ to form the representation of dataset $D_{kj}$, thus avoiding redundant gradient calculations for each data point in datasets $D_k$ and $D_j$. In summary, our method only requires calculating the gradient for each data owner, resulting in a time complexity of $O(n)$, which is crucial for the practical application.

Table 1: Comparison of estimated data value across different **course-grained** domains for NESTLE.

| Domain | Target | Finance | | | Health | | | Law | | |
|---|---|---|---|---|---|---|---|---|---|---|
| | Source | Finance | Health | Law | Finance | Health | Law | Finance | Health | Law |
| Llama2-7B | | **0.708** | 0.160 | 0.132 | 0.126 | **0.551** | 0.323 | 0.250 | 0.547 | **0.203** |
| ChatGLM3-6B | | **0.527** | 0.192 | 0.281 | 0.254 | **0.540** | 0.206 | 0.294 | 0.182 | **0.525** |
| Qwen1.5-7B | | **0.619** | 0.114 | 0.268 | 0.134 | **0.664** | 0.202 | 0.192 | 0.155 | **0.653** |

Table 2: Comparison of estimated data value across different **fine-grained** domains for NESTLE.

| Domain | Target | Consult | | | TCM | | | Medicine | | |
|---|---|---|---|---|---|---|---|---|---|---|
| | Source | Consult | TCM | Medicine | Consult | TCM | Medicine | Consult | TCM | Medicine |
| Llama2-7B | | **0.611** | 0.229 | 0.160 | 0.275 | **0.561** | 0.164 | 0.188 | 0.1880 | **0.623** |
| ChatGLM3-6B | | **0.436** | 0.294 | 0.271 | 0.241 | **0.574** | 0.195 | 0.237 | 0.213 | **0.550** |
| Qwen1.5-7B | | **0.511** | 0.259 | 0.230 | 0.221 | **0.600** | 0.179 | 0.201 | 0.206 | **0.593** |

# 4 EXPERIMENTS

In this section, we provide the experimental results to verify the effectiveness and robustness of our proposed data valuation framework NESTLE. More results can be found in Appendix.

## 4.1 EXPERIMENTAL SETTINGS

**Evaluation Protocols.**  As mentioned in Section 3.3, a valuation framework needs to estimate the multi-domain value distribution for a single data provider and dynamically adjust values for coexisting multiple providers. Hence, our evaluation scenarios are categorized into single-source and multi-source scenarios. (i)- **Single-source** evaluation primarily assesses whether the framework accurately estimates the value distribution for a single data provider across different target domains. (ii)- **Multi-source** evaluation primarily assesses whether the framework can perform unbiased and fair dynamic adjustments for multiple coexisting data providers within a specific domain.

**Datasets and LLMs.**  Our proposed framework is evaluated on a variety of datasets. For the single-source setting, the evaluation is conducted at two different cross-domain granularities: coarse-grained and fine-grained. (i)-For coarse-grained cross-domain data, we adopt the fields of `Finance`, `Healthcare`, and `Law`. (ii)-For fine-grained cross-domain data, we adopt three sub-fields under the healthcare domain: `Consult` Zhu (2023), `Medicine`, and `TCM` (traditional Chinese medicine). For the multi-source setting, the evaluation is focused on the consult domains, with multiple data providers possessing different to-be-valuated data in this consult field. For each of these domains and subdomains, we randomly divide 1000 samples to form the support set $D_{sup}$, and the to-be-valuated data $D_{valu}$ are sampled from the remaining samples. Our framework is architecture-agnostic and compatible with any open-source LLMs, hence we mainly select some representative ones for evaluation, including `Llama2-7B`, `Qwen1.5-7B`, and `ChatGLM3-6B`. We also included explorations on larger models of the Qwen series LLMs, including `Qwen1.5-14B` and `Qwen1.5-32B`. The complete statistics can be found in Appendix.

**Evaluation Metrics.**  Under the single-source setting, we primarily consider the consistency between the estimated data value distribution and the true data distribution. We tested the data value from different sources across various target domains under both coarse-grained and fine-grained domain distributions. Under the multi-source setting, our main focus is on the alignment between the data value distribution of various coexisting providers and the corresponding ground truth value. The ground truth of the data value distribution is represented by the traditional Shapley value (SV), which is discussed in Eq. (7) and reveals the marginal performance improvement. We employ commonly-adopted matching-based metrics like BLEU, ROUGE-1, ROUGE-2, and ROUGE-L as utility functions $U$. The detailed experimental cases are provided later in Section 4.2.

Table 3: Performance comparisons under multi-source evaluations for cases 1 to 4.

| Multi-Source Cooperative Setting | | Ground-truth Shapley value | | | | LOO | FedCE | NESTLE |
|---|---|---|---|---|---|---|---|---|
| | | BLEU-4 | ROUGE-1 | ROUGE-2 | ROUGE-L | | | |
| Case 1 | Provider 1 | 0.3127 | 0.3204 | 0.3225 | 0.3217 | 0.2888 | 0.2817 | 0.2644 |
| | Provider 2 | 0.3392 | 0.3369 | 0.3321 | 0.3338 | 0.2047 | 0.3523 | 0.3428 |
| | Provider 3 | 0.3479 | 0.3426 | 0.3453 | 0.3440 | 0.5066 | 0.3661 | 0.3927 |
| Case 2 | Provider 1 | 0.3043 | 0.2974 | 0.2972 | 0.2969 | 0.3131 | 0.3132 | 0.2508 |
| | Provider 2 | 0.3427 | 0.3450 | 0.3309 | 0.3356 | 0.3696 | 0.3333 | 0.3469 |
| | Provider 3 | 0.3528 | 0.3612 | 0.3718 | 0.3674 | 0.3171 | 0.3535 | 0.4000 |
| Case 3 | Provider 1 | 0.3136 | 0.2897 | 0.2736 | 0.2983 | 0.3818 | 0.3165 | 0.3048 |
| | Provider 2 | 0.3225 | 0.3380 | 0.3409 | 0.3389 | 0.1541 | 0.3231 | 0.3429 |
| | Provider 3 | 0.3608 | 0.3733 | 0.3854 | 0.3628 | 0.4640 | 0.3603 | 0.3523 |
| Case 4 | Provider 1 | 0.3597 | 0.3524 | 0.3478 | 0.3394 | 1.0 | 0.6718 | 0.3838 |
| | Provider 2 | 0.3201 | 0.3237 | 0.3260 | 0.3302 | 0.0 | 0.1641 | 0.3080 |
| | Provider 3 | 0.3201 | 0.3237 | 0.3260 | 0.3302 | 0.0 | 0.1641 | 0.3080 |

**Implementation Details.** The low-rank adaptation Hu et al. (2022) is adopted when calculating the gradients in our framework and fine-tuning models in the baselines for calculating Shapley values. The LoRA rank $r$ is set as 8 and the LoRA alpha $\alpha$ is set as 32. All the adopted open-sourced LLMs leverage the instruct/chat version instead of the base version. All the experiments are conducted with 4×A100-80G GPUs. More implementation details can be found in Appendix.

## 4.2 EXPERIMENTAL RESULTS

**Cross-Domain Valuation of Single-Source Evaluation.** To evaluate the valuation ability across different target domains for our framework. We conduct extensive experiments under the coarse-grained and fine-grained setup. The experimental results are shown in Table 2. It can be observed that the estimated data value varies significantly across the three domains. In different target domains, the changes in value estimation accurately reflect the structure of data sources, and the data not belonging to the target domain is always assigned lower scores. Hence, it can be employed for targeted data filtering by setting up a directional support dataset $D_{sup}$ for a specific domain for fine-tuning LLMs in specific downstream tasks.

**Dynamic Adjustment in Multi-Source Evaluation.** More importantly, in real-world evaluation scenarios where multiple data providers coexist, the data valuation frameworks need to dynamically adjust to different collaborative contexts. To evaluate the adaptability of our proposed framework across various scenarios, we concentrate on the medical consult subdomain, simulating multiple cooperative cases based on the redundancy of data owned by different providers. Each case involves three sources (data provider) which are denoted as $P_1$, $P_2$ and $P_3$. The details are as follows:

- Case 1: There is no data overlap among the different providers: $P_1$ possesses 20% of the data, $P_2$ possesses 33.3%, and $P_3$ holds the remaining 46.7%. ($P_1$:$P_2$:$P_3 = 3k : 5k : 7k$)

- Case 2: There is a partial overlap between $P_2$ and $P_3$. $P_1$ possesses 20% of the data, $P_2$ possesses 40%, and $P_3$ holds 60%. The overlapping section makes up 20% of the total data. ($P_1 : P_2 : P_3 = 3k : 6k : 9k$, with $3k$ overlap between $P_2$ and $P_3$)

- Case 3: There is a partial overlap between $P_1$ and $P_2$, and another partial overlap between $P_2$ and $P_3$ (The two overlapping parts are independent of each other.). $P_1$ possesses 33.3% of the data, $P_2$ possesses 50%, and $P_3$ holds 50%. with a 16.67% overlap between $P_1$ and $P_2$, and approximately 16.67% overlap between $P_2$ and $P_3$. ($P_1 : P_2 : P_3 = 6k : 9k : 9k$, with $3k$ overlap between $P_1$ and $P_2$, another $3k$ overlap between $P_2$ and $P_3$)

- Case 4: There is a complete data overlap between $P_2$ and $P_3$. Each of the three providers holds 50% of the total data, and the data section held by $P_2$ and $P_3$ is exactly the same. ($P_1 : P_2 : P_3 = 3k : 3k : 3k$, with $3k$ overlap between $P_2$ and $P_3$)

Table 4: Performance comparisons with different numbers of data providers $N = 3, 4, 5$.

| Number of Data Providers | | Ground-truth Shapley value | | | | NESTLE |
|---|---|---|---|---|---|---|
| | | BLEU-4 | ROUGE-1 | ROUGE-2 | ROUGE-L | |
| N = 3 | $P_1 = 3k$ (16.7%) | 0.2957 | 0.3150 | 0.3050 | 0.3111 | 0.2664 |
| | $P_2 = 6k$ (33.3%) | 0.3433 | 0.3346 | 0.3164 | 0.3253 | 0.3403 |
| | $P_3 = 9k$ (50.0%) | 0.3608 | 0.3504 | 0.3786 | 0.3636 | 0.3931 |
| N = 4 | $P_1 = 3k$ (10%) | 0.2217 | 0.2320 | 0.2105 | 0.2289 | 0.1911 |
| | $P_2 = 6k$ (20%) | 0.2562 | 0.2437 | 0.2277 | 0.2374 | 0.2402 |
| | $P_3 = 9k$ (30%) | 0.2573 | 0.2569 | 0.2807 | 0.2557 | 0.2742 |
| | $P_4 = 12k$ (40%) | 0.2648 | 0.2674 | 0.2811 | 0.2780 | 0.2944 |
| N = 5 | $P_1 = 3k$ (6.7%) | 0.1759 | 0.1770 | 0.1649 | 0.1790 | 0.1489 |
| | $P_2 = 6k$ (13.3%) | 0.1978 | 0.1831 | 0.1667 | 0.1814 | 0.1842 |
| | $P_3 = 9k$ (20.0%) | 0.2035 | 0.1988 | 0.2097 | 0.2072 | 0.2092 |
| | $P_4 = 12k$ (26.7%) | 0.2083 | 0.2140 | 0.2223 | 0.2124 | 0.237 |
| | $P_5 = 15k$ (33.3%) | 0.2142 | 0.2271 | 0.2364 | 0.2200 | 0.2338 |

For comparison, the ground-truth Shapley value (SV) with different utility functions (BLEU-4, ROUGE-1, ROUGE-2, and ROUGE-L) for each provider in these four cases is calculated using the `Llama2-7B` model. The results are normalized and shown in Table 3. It can be observed that even though different utility functions yield slightly varying numerical Shapley values, they share the same valuation trend, with very small numerical differences. This confirms the stability and reliability of the Shapley value as a ground truth. The results also demonstrate that our method can handle scenarios with varying degrees of data overlap. The estimated data value maintains the same order as the ground truth. The valuation results also fulfill the properties of *Strict Monotonicity* and *Symmetry* mentioned in Section 3.1.

Furthermore, We compared our framework with other valuation baselines, including the Leave-One-Out (LOO) Jia et al. (2019) and FedCE Jiang et al. (2023b) methods. As shown in Table 3, we found that other baselines exhibit undesired estimation in certain overlapping scenarios. For example, in Case 4 where $P_2$ and $P_3$ possess the same data section. According to the characteristics of the LOO method, the estimated data values of both $P_2$ and $P_3$ are 0, as they can replace each other in the LOO setting. This result is evidently inaccurate as the values of $P_2$ and $P_3$ should face penalties, but they should not be reduced to zero. On the other hand, FedCE can achieve valuation results closer to the ground truth SV than LOO. However, FedCE shows severe estimation bias in fully overlapping scenarios of Case 4. Further, the dependency on local and global cooperation in federated training restricts the adaptability and transferability of FedCE. When a new data provider joins the collaborative setting, recomputation for all clients (data providers) is needed to reach a new balance, resulting in substantial additional computation that hinders FedCE's flexible transferability. Compared to these methods, our approach can accurately estimate the data value in the collaborative multi-source setting. Additionally, due to the additive property of the gradients, Our approach allows for cost-effective adaptation to scenarios where new data providers participate.

## 4.3 ANALYTICAL STUDIES

**Exploration of the Valuation Cost.** We conducted an analysis of time consumption for various valuation methods. We test the scenario with 3 data providers on Llama2-7B. Each provider possesses 3k non-overlapping data samples. The time costs are displayed in Table 5, the time of 240min×7 means there need 7 sets of running in total, which can be executed concurrently. Each of the running takes approximately 240 minutes on average. It can be observed that our proposed framework is much more efficient than the other data valuation baselines, e.g., it costs only 1.25% of time compared to the ground-truth Shapley Value baseline.

Table 5: The valuation time.

| Valuation Method | Total Time |
|---|---|
| Shapley Value | 240min × 7 |
| LOO | 240min × 4 |
| FedCE | 60min |
| NESTLE (ours) | 7min × 3 |

**Impact of More Sources of Data Providers.** In a multi-source setting, we further explored data valuation experiments when more collaborative data providers are involved, including cases with $N = 3, 4, 5$. The specific data provider configurations and experimental results are shown in Table 4. Notably, in these settings, there is no data overlap among different data providers. It can be observed that when more sources of data providers are incorporated, our framework remains robust and its valuation results remain consistent with the ground-truth Shapley value. While our method is much more efficient than the vanilla Shapley value.

**Impact of Integrating Trained Data into $D_{valu}$.** To further verify the robustness of our method, we explored a variant scenario where the to-be-valuated data is adulterated with a portion of data that has been used for the fine-tuning of the LLM $M_\theta$. Specifically, the proportions of the mixed fine-tuned data are 0%, 25%, 50%, 75% and 100% respectively. As shown in Table 6, with the total data volume unchanged, the overall estimated value of the data significantly declines as the proportion of the already-trained data grows. Such results exactly match the expectations, since in practice, data that the LLM has already been trained on tends to

Table 6: Estimated value with different ratios of already-trained data.

| Ratio of Already Trained Data | Estimated Data Value |
|---|---|
| 0% | 0.5897 |
| 25% | 0.5751 |
| 50% | 0.5605 |
| 75% | 0.5517 |
| 100% | 0.5374 |

provide less benefit to the same LLM, as its pattern has been previously learned. Such influence is captured in our method through a reduction in the gradient magnitude of the evaluated data $D_{valu}$.

**Analysis of the Marginal Effects.** In the main results section, we discussed the performance of our valuation framework across different target domains and in scenarios with multiple data providers. Here we further investigate the trend in total data value at the sample-wise level with varying sample quantities. As shown in Figure 1, we conduct experiments on the 7B, 14B, and 32B models of the Qwen-1.5 series across the spectrum of 0 to 200k samples. Notably, Due to the non-comparable nature of value magnitudes across different LLMs, we have normalized the different value curves to a uniform scale between 0 and 1. It can be observed that, as the number of samples increases, the data value initially grows rapidly and then gradually stabilizes, reflecting

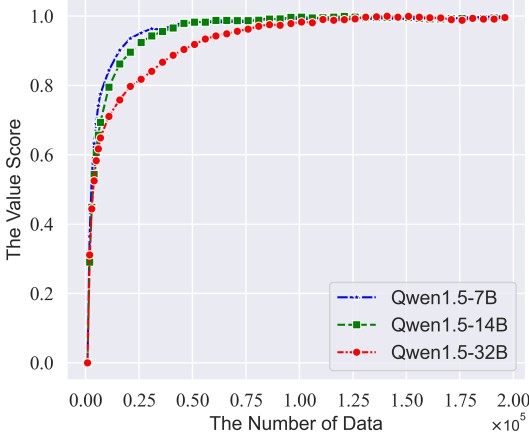

Figure 1: Value score curves on different LLMs

a diminishing marginal effect. This trend is present across LLMs of different sizes, though the saturation point varies with each size. As the model size increases from 7B to 32B, the inflection point for total value saturation corresponds to larger data volumes. This indicates that larger models require more data to achieve optimal fitting in a given scenario. Such empirical findings align with the scaling laws of instruction tuning of LLMs.

## 5 CONCLUSION

In this paper, we propose a gradient-based data valuation framework, NESTLE, for different downstream fine-tuning tasks of LLMs. We first find that traditional data valuation methods typically rely on cumbersome re-training and can be distorted. Thus we present our training-free valuation algorithm based on gradient tracing to accurately and efficiently estimate the data value distribution across different target domains. To further address the potential dynamic adjustments in multi-source scenarios, we combine our grading tracing mechanisms with Shapley value (SV) theory with the additivity of gradients, effectively evaluating the contribution of each data provider. Extensive experiments demonstrate that our accelerated Shapley-based gradient estimation can accurately adjust the data value distribution, while requiring very little calculation cost. Future endeavors could explore the connection between value estimation and data scaling law.

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
