# NESTLE: An Efficient and Robust Data Valuation Framework for Large Language Models

## A    Appendix

### A.1    Dataset Description

We evaluated the effectiveness of the method using the `Finance` [1], `Healthcare` [2], `Law` [3], `Consult`[4], `Medicine` [5], and `TCM` [6] datasets. The Finance dataset contains $53.9k$ financial records, Healthcare contains $112k$ medical records, Law contains $9.2k$ legal records, Consult contains $549k$ medical consultation records, Medicine contains $20k$ Western medicine records, and TCM contains $113k$ traditional Chinese medicine records. We randomly selected 1000 entries as the target domain dataset.

### A.2    Contribution Analysis in Cooperative Games

In order to extensively validate the efficacy of our model, we also estimated the contributions of participants in a cooperative game using `Qwen1.5` and `ChatGLM3`. The experimental results are shown in Table 1 and Table 2. It can be observed that, for other models, the ranking of participant contributions is consistent with other performance metrics. Notably, due to the variability in metric evaluations, it is notable that evaluations on `Qwen1.5` exhibit discrepancies specifically in the ROUGE-1, which also demonstrates the superior robustness of our method.

Table 1: Evaluation of participant contributions in cooperative games using the `ChatGLM3-6B` model. ($P_i$ denotes the $i$ provider.)

| Mulri-Source Cooperative Setting | | Ground-truth Shapley value | | | | OURS |
|---|---|---|---|---|---|---|
| | | BLEU-4 | ROUGE-1 | ROUGE-2 | ROUGE-L | |
| $C_1$ | $P_1$ | 0.2963 | 0.3043 | 0.2849 | 0.3086 | 0.3056 |
| | $P_2$ | 0.3250 | 0.3312 | 0.3314 | 0.3217 | 0.3324 |
| | $P_3$ | 0.3785 | 0.3644 | 0.3835 | 0.3696 | 0.3619 |
| $C_2$ | $P_1$ | 0.3203 | 0.3242 | 0.3288 | 0.3220 | 0.3294 |
| | $P_2$ | 0.3308 | 0.3266 | 0.3254 | 0.3286 | 0.3232 |
| | $P_3$ | 0.3488 | 0.3491 | 0.3456 | 0.3492 | 0.3473 |
| $C_3$ | $P_1$ | 0.3255 | 0.3105 | 0.3186 | 0.3101 | 0.3111 |
| | $P_2$ | 0.3308 | 0.3334 | 0.3281 | 0.3377 | 0.3276 |
| | $P_3$ | 0.3435 | 0.3561 | 0.3533 | 0.3521 | 0.3613 |
| $C_4$ | $P_1$ | 0.3634 | 0.3679 | 0.3508 | 0.3805 | 0.3544 |
| | $P_2$ | 0.3183 | 0.3160 | 0.3245 | 0.3097 | 0.3228 |
| | $P_3$ | 0.3183 | 0.3160 | 0.3245 | 0.3097 | 0.3228 |

[1] https://huggingface.co/datasets/4DR1455/finance_questions
[2] https://huggingface.co/datasets/wangrongsheng/HealthCareMagic-100k-en
[3] https://huggingface.co/datasets/Alignment-Lab-AI/Lawyer-Instruct
[4] https://huggingface.co/datasets/michaelwzhu/ChatMed_Consult_Dataset
[5] https://github.com/CMKRG/QiZhenGPT
[6] https://huggingface.co/datasets/michaelwzhu/ShenNong_TCM_Dataset

Table 2: Evaluation of participant contributions in cooperative games using the `Qwen1.5-7B` model. ($P_i$ denotes the $i$ provider.)

| Mulri-Source Cooperative Setting | | Ground-truth Shapley value | | | | OURS |
|---|---|---|---|---|---|---|
| | | BLEU-4 | ROUGE-1 | ROUGE-2 | ROUGE-L | |
| $C_1$ | $P_1$ | 0.3250 | 0.3233 | 0.3215 | 0.3250 | 0.2973 |
| | $P_2$ | 0.3321 | 0.3367 | 0.3334 | 0.3309 | 0.3404 |
| | $P_3$ | 0.3428 | 0.3398 | 0.3450 | 0.3438 | 0.3623 |
| $C_2$ | $P_1$ | 0.3129 | 0.3131 | 0.3075 | 0.3186 | 0.2910 |
| | $P_2$ | 0.3333 | 0.3138 | 0.3385 | 0.3362 | 0.3409 |
| | $P_3$ | 0.3536 | 0.3729 | 0.3539 | 0.3450 | 0.3681 |
| $C_3$ | $P_1$ | 0.3282 | 0.3245 | 0.3219 | 0.3260 | 0.3173 |
| | $P_2$ | 0.3342 | 0.3271 | 0.3313 | 0.3258 | 0.3389 |
| | $P_3$ | 0.3375 | 0.3483 | 0.3467 | 0.3482 | 0.3436 |
| $C_4$ | $P_1$ | 0.3341 | 0.3297 | 0.3432 | 0.3394 | 0.3645 |
| | $P_2$ | 0.3329 | 0.3351 | 0.3284 | 0.3302 | 0.3177 |
| | $P_3$ | 0.3329 | 0.3351 | 0.3284 | 0.3302 | 0.3177 |