# OpenReview forum: "NESTLE: An Efficient and Robust Data Valuation Framework for Large Language Models"
_ICLR.cc/2025/Conference — Submitted to ICLR 2025_

### Official Review · Reviewer_sfUx · 2024-10-22

**Soundness:** 3
**Presentation:** 3
**Contribution:** 2
**Rating:** 5
**Confidence:** 4

**Summary:**

The paper proposes NESTLE, a framework for conducting data valuation using its influence on the change of the loss. This results in calculating the gradient inner product between the gradient on the dataset and the gradient of the particular data point. To resolve the comupational issues, the authors propose to project the gradients to a lower dimensional space. For dealing with multiple data provider, the authors propose to use concepts in Shapley value to define the marginal gain in each data source. Experiments are conducted to showcase the effectiveness the proposed method.

**Strengths:**

The paper is well presented. The idea of using gradient inner product to evaluate the influence of the training data is clear and intuitive. To deal with the multi-source setting, using Shapley value to think through this is interesting.

**Weaknesses:**

My main concern is that the proposed way for calculating the value/influence of a data point in model training (in the single source setting) is almost identical to that proposed in LESS: https://arxiv.org/pdf/2402.04333. For multi-source provider case, the adjustment using the marginal utility of adding a data source is interesting, but unclear to me whether it suffices to be enough contribution.

As it was motivated at the beginning of the paper, one cares about data valuation because of training needs. However, in the experiment section, the evaluation metrics for the proposed methods seem to be sentence statistic based, lacking the connection to model training. Thus, it is hard to assess whether the proposed method will be effective in training.

**Questions:**

See weakness above. Could you comment on given that there are existing influence-based data selection work (like LESS), how should one value the proposed framework? In addition, if BLEU scores can be used to assess the quality of data valuation, then perhaps one can use it for data valuation directly?

---

> ### Author Response · Authors · 2024-11-25
> **Response to Reviewer sfUx (Part 1/2)**
>
> Thanks very much for the time and effort in reviewing our paper. We hope the following responses can clarify your confusion.
>
> **Q1: My main concern is that the proposed way for calculating the value/influence of a data point in model training (in the single source setting) is almost identical to that proposed in LESS:** [**https://arxiv.org/pdf/2402.04333**](https://arxiv.org/pdf/2402.04333)**. For multi-source provider case, the adjustment using the marginal utility of adding a data source is interesting, but unclear to me whether it suffices to be enough contribution.**
>
> **A1**:  Thanks for the question. Our proposed method is fundamentally different from these data filtering methods based on influence functions.
>
> ​	(a) **Task and Application**:  These data selection methods is proposed for data selection, which aims to select the most influential subsets during the training process to enhance the final training outcome and are typically applied in single-domain and single-source scenarios. In contrast, our method is focused on directly outputting the value distribution of data from various sources across different target domains, offering a more general setting that caters to multi-domain and multi-data-source scenarios. Besides data selection, our method can also be applied to data pricing, data trading, and other related tasks.
>
> ​	(b) **Data Flow**:  These data selection methods, like LESS, typically require a multi-stage training process that involves three full stages of warm-up, data filtering, and training, which is cost-intensive. Our approach, on the other hand, does not require training and updating the model parameters but rather involves efficient computation of the gradient of the valuated data.
>
> Our approach originally takes inspiration from the gradient-inner-product-based estimation of data influence and propose a comprehensive data valuation framework on top of that. However, such gradient-based estimation techniques did not originate from the aforementioned data selection methods. Instead, they were initially derived from theoretical studies related to influence functions [1] and have since seen extensive development and variations in fields like data distillation [2], data attribution [3], and others. Our NESTLE framework, which is built upon such gradient-based data influence estimations, presents a comprehensive data valuation framework for the novel task of assessing data value in LLMs, considering various data sources and different target domains.
>
> [1] Koh, Pang Wei, and Percy Liang. "Understanding black-box predictions via influence functions." 2017.
>
> [2] Loo, Noel, et al. "Dataset distillation with convexified implicit gradients." *International Conference on Machine Learning*. 2023.
>
> [3] Singla, Vasu, et al. "A Simple and Efficient Baseline for Data Attribution on Images." 2023.
>
>
>
> **Q2: As it was motivated at the beginning of the paper, one cares about data valuation because of training needs. However, in the experiment section, the evaluation metrics for the proposed methods seem to be sentence statistic based, lacking the connection to model training. Thus, it is hard to assess whether the proposed method will be effective in training.**
>
> **A2**: (1) In the multi-source data comparison presented in Table 3, we compare our data valuation results with the ground-truth Shapley values in a complex, real-world setting. The ground-truth Shapley values in Table 3 are obtained by using the data from different clients to train the LLM and then calculating the relevant downstream metrics. We have found that our data valuation method shows a high level of consistency with the actual Shapley values.
>
> (2) We further provide more direct results in single-source scenarios. From the perspective of data selection, similar to setting in Table 2, the target domain is set to Consult and we select a 4000-sample subset from the original 15000-sample dataset with 3 different strategies: (a)-**Maximum Selection**, which selects samples with the largest data scores. (b)-**Minimum Selection**, which selects samples with the smallest data scores. (c)-**Random Selection**, which selects samples in a random manner. The filtered subset is then employed to fine-tune a ChatGLM3-6B model and test its downstream performance on target domains. The experimental results are shown below, indicating that data with larger data scores leads to better performance in downstream training.
>
> |                   | BLEU-4  | ROUGE-1 | ROUGE-2 | ROUGE-L |
> | ----------------- | ------- | ------- | ------- | ------- |
> | Maximum Selection | 13.2779 | 36.6102 | 12.9326 | 26.7862 |
> | Random Selection  | 13.0454 | 36.5104 | 12.6523 | 26.5085 |
> | Minimum Selection | 12.2375 | 35.7346 | 12.2955 | 25.9136 |
>
> In addition, as mentioned in Q1, our data valuation framework is a general task scenario that can serve various purposes such as data trading and data pricing. Downstream training is just one of these potential applications.

---

> ### Author Response · Authors · 2024-11-25
> **Response to Reviewer sfUx (Part 2/2)**
>
> **Q3: See weakness above. Could you comment on given that there are existing influence-based data selection work (like LESS), how should one value the proposed framework?**
>
> **A3**: As stated in Q1, our data valuation framework serves as a general task scenario, and data selection and training represent only one of the downstream application areas. Therefore, our main evaluation criterion is the consistency between the estimated values and the ground-truth Shapley values across different scenarios. For specific applications, such as data selection and model training, we can further examine the consistency between the data valuation/data selection methods and the actual outcomes. We will add more discussion on this aspect in the revision.
>
> **Q4: In addition, if BLEU scores can be used to assess the quality of data valuation, then perhaps one can use it for data valuation directly?**
>
> **A4**: As mentioned in our original submission, one of the most straightforward approaches to data valuation is to directly employ the to-be-valuated-data for model training and compare the metrics from downstream tasks, such as BLEU, as the direct standard for data value (that is, data with higher value corresponds to greater gains in downstream metrics). However, this method has some significant drawbacks: (a)-it is costly, requiring a complete training and evaluation process; (b)-it is unstable and unreliable, as it is heavily influenced by initialization and hyperparameters, and different metrics can even conflict with each other (e.g., metric BLEU may indicate that the data from provider 1 is more valuable, while metric ROUGE may suggest that the data from provider 2 is more valuable). To address this, in our paper, we propose a low-cost gradient-tracing-based method, achieving precise and efficient data valuation through a fast Shapley-based contribution measurement approach.

---

> ### Author Response · Authors · 2024-12-02
> **Sincerely Looking Forward to Your Feedback**
>
> Dear reviewer sfUx, thank you once more for your valuable input. With the discussion period nearing its end, if you have any additional questions about our manuscript, we would be delighted to address them. Thank you very much for your attention to our work.

---

### Official Review · Reviewer_eHfe · 2024-10-22

**Soundness:** 3
**Presentation:** 3
**Contribution:** 3
**Rating:** 8
**Confidence:** 2

**Summary:**

This paper introduces NESTLE, which is a framework that:
- tackles estimating the value of (or, as I interpret it – the information present in) a data point with respect to (wrt) a pool of points
- represents the value of data using model gradients
- can change the data value efficiently wrt multiple pools of data

**Strengths:**

- Interesting problem statement
- Addresses inherent weaknesses by (1) reducing gradient caching costs and (2) adapting gradient computation to the Adam optimizer
- Added an analysis for runtime, which makes the efficiency claims stronger
- Figure 1 provides cool insights into the delta of value added to datasets as they get larger, demonstrating that large datasets are not necessary for fine-tuning models

**Weaknesses:**

- Comparison with multiple data providers (Table 4) is not complete. It would be better if the value estimation were also done on other baselines.
- A more thorough analysis on previous works using model gradients for estimating data value would be great, and motivate the paper’s solution much better. A few papers that come to mind are:
1. S. K. Choe et al: “What is Your Data Worth to GPT? LLM-Scale Data Valuation with Influence Functions”
2. M. Xia et al: “LESS: Selecting Influential Data for Targeted Instruction Tuning”
- A nitpicky thing: wrap your citations in parentheses (maybe use `\citep` in latex).

**Questions:**

I just have one question on top of the suggestions mentioned above: the problem you are trying to solve sounds a lot like active learning. Do you have any experiments in which you compute the value of data, treat the high-valued data as an information-rich subset of the full data, and fine-tune a model on the subset? What are the results?

---

> ### Author Response · Authors · 2024-11-25
> **Response to Reviewer eHfe**
>
> Thank you for your valuable time and review, and we have made great efforts to address all these concerns below. We are happy to provide additional explanations if needed.
>
>
>
> **Q1: Comparison with multiple data providers (Table 4) is not complete. It would be better if the value estimation were also done on other baselines.**
>
> **A1**: Thanks for your suggestions. Our main comparison results across different baselines are already displayed in Table 3 to demonstrate the advantages of our approach. The goal of Table 4 is to demonstrate that our method is also suitable for scenarios with more data providers.
>
> We have further supplemented the experimental comparisons with other baselines in Table 4, as shown below.
>
> | Provider Number |  LOO    | FedCE  | NESTLE (ours) |
> | ---- | ------ | ------ | ------------- |
> | N=3, P-1  | 0.2888 | 0.2817 | 0.2644        |
> | N=3, P-2  | 0.2047 | 0.3523 | 0.3428        |
> | N=3, P-3  | 0.5066 | 0.3661 | 0.3927        |
> | N=4, P-1 | 0.1683 | 0.2031 | 0.1911        |
> | N=4, P-2| 0.2399 | 0.2347 | 0.2402        |
> | N=4, P-3| 0.2838 | 0.2730 | 0.2742        |
> | N=4, P-4| 0.3080 | 0.2891 | 0.2944        |
> | N=5, P-1| 0.1686 | 0.1609 | 0.1489        |
> | N=5, P-2| 0.2011 | 0.1773 | 0.1842        |
> | N=5, P-3| 0.2048 | 0.2027 | 0.2092        |
> | N=5, P-4| 0.2150 | 0.2203 | 0.2270        |
> | N=5, P-5| 0.2104 | 0.2388 | 0.2338        |
>
>
>
>
>
> **Q2: A more thorough analysis on previous works using model gradients for estimating data value would be great, and motivate the paper’s solution much better. A few papers that come to mind are:**
>
> **[1]. S. K. Choe et al: “What is Your Data Worth to GPT? LLM-Scale Data Valuation with Influence Functions”**
>
> **[2]. M. Xia et al: “LESS: Selecting Influential Data for Targeted Instruction Tuning”**
>
> **A2**: Thank you for your suggestions. We will include a discussion on these related works in our revision. Additionally, we want to clarify that our proposed method is fundamentally different from the previous influence-function-based methods in [1] [2].
>
> (a) **Task and Application**:  These influence-function-based methods is proposed for data selection, which aims select the most influential subsets during the training process to enhance the final training outcome and are typically applied in single-domain and single-source scenarios. In contrast, our method is focused on directly outputting the value distribution of data from various sources across different target domains, offering a more general setting that caters to multi-domain and multi-data-source scenarios. Besides data selection, our method can also be applied in data pricing, data trading, and other related tasks.
>
> (b) **Data Flow**:  These influence-function-based methods typically require a multi-stage training process that involves three full stages of warm-up, data filtering, and training, which is cost-intensive. Our approach, on the other hand, does not require training and updating the model parameters but rather only involves efficient computation of the gradient of the valuated data.
>
>
>
> **Q3: I just have one question on top of the suggestions mentioned above: the problem you are trying to solve sounds a lot like active learning. Do you have any experiments in which you compute the value of data, treat the high-valued data as an information-rich subset of the full data, and fine-tune a model on the subset? What are the results?**
>
> **A3**: Thank you for your question. Our method is primarily designed for a general task scenario of evaluating data value across multiple data sources and target domains, rather than being limited to subfield of the data selection and active learning.
>
> Here we have further provided experimental results for data subset selection. Similar to setting in Table 2, the target domain is set to Consult and we select a 4000-sample subset from the original 15000-sample dataset with 3 different strategies: (1) **Maximum Selection**, which selects samples with the largest data scores. (2) **Minimum Selection**, which selects samples with the smallest data scores. (3) **Random Selection**, which selects samples in a random manner. The filtered subset is then employed to fine-tune a ChatGLM3-6B model and test its downstream performance on target domains. The results, shown below, indicate that the subset with the highest data score also results in better downstream training performance. Such consistency also confirms the effectiveness of our method in active learning scenarios related to data selection.
>
> |                   | BLEU-4  | ROUGE-1 | ROUGE-2 | ROUGE-L |
> | ----------------- | ------- | ------- | ------- | ------- |
> | Maximum Selection | 13.2779 | 36.6102 | 12.9326 | 26.7862 |
> | Random Selection  | 13.0454 | 36.5104 | 12.6523 | 26.5085 |
> | Minimum Selection | 12.2375 | 35.7346 | 12.2955 | 25.9136 |
>
>
>
> **Q4: A nitpicky thing: wrap your citations in parentheses (maybe use** `**\citep**` **in latex).**
>
> **A4**: We will double-check and fix it in our revision!

---

> > ### Comment · Reviewer_eHfe · 2024-11-25
> >
> > Thank you for taking the time to provide a detailed responses. All my concerns are addressed. I've increased my rating.

---

> > > ### Author Response · Authors · 2024-11-26
> > > **Thanks for the feedback**
> > >
> > > Thanks for your reply! We sincerely thank you for your valuable time and support on our paper!

---

### Official Review · Reviewer_Pz1e · 2024-11-03

**Soundness:** 1
**Presentation:** 1
**Contribution:** 1
**Rating:** 3
**Confidence:** 3

**Summary:**

The goal of the paper is to calculate fair values of the data that comes from different providers. The paper first describes the two broad approaches, that is, the influence function based and Shapley values based approaches. The paper combines aspects from the two approaches. It first uses the gradient information from the influence based procedures to compute the value of the data. This computation is based on the Taylor series expansion and seems to be heavily inspired by existing influence functions (e..g., Koh and Liang). Then noting that multiple providers may have the same utility, leverages SVs to compute the fair valuation of the data. The paper leverages the "additivity property of gradients" to speed up the computation of Shapley values. The result is a method that can compute valuation of fine-tuning data of pretrained models while also accounting for redundancy of data provided by different sources.

**Strengths:**

1. Valuation of fine-tuning data is an important problem and will likely be highly relevant given that fine-tuning LLMs with domain specific data is a common use-case.

2. The paper sets up the problem and the context quite nicely. It also explains the evaluation desiderata in Section 3.1 very well.

**Weaknesses:**

The main points where the paper can be significantly improved are (i) lack of novelty, (ii) insufficiently justified assumptions and (iii) unsuitability of the experimental setup. See below for details.

1. The technical novelty of the paper is somewhat limited. The Taylor expansion which underpins the gradient mechanism is already quite well-known in the influence function literature. The usage of Shapley values is also quite common in data valuation literature. The contribution of the paper seems to be a fast way to compute them but the paper does not provide sufficient detail on that computation (more on this point below).

2. Eq. 5: Given that the loss is a mere proxy for the task accuracy, it is not clear how reliable the result of “summing up this formula in all the iterations in which $s_i$ was used to update the parameters” is. For instance, according to [Guo et al](https://arxiv.org/pdf/1706.04599), the accuracy and loss do not always go hand in hand. Does this difference between loss and accuracy, and the non-convexity of deep models have any effect on the reliability of the influence estimate? I think the paper would be greatly strengthened by adding some clarifications here.

3. Eq. 6: The insight that the other parameters of the Adam optimizer should also be considered makes sense on a high level. However, the equation is discussed in a very standalone manner and leads to more questions than answers. Why the first two moments? What is meant by “calibration”? How does the equation generalize to related optimizers like AdamW and SGD+Momentum?

4. The treatment of “domains” in the paper is a bit hand-wavy. Are domains so easy to define in the era of LLMs where the number of tasks one could perform with a LLM is virtually unlimited. Similarly, the update strategy seems to implicitly assume that we have similar number of data points from each domain (the way the losses for each data points are summed up over the iterations). Some domains, however, may contribute much less data and may have a higher loss. Does the paper suggest to train on these domains for longer? How does this impact the loss summation strategy?

5. The text below Eq. 7 is very difficult to parse. The phrase “the required number of training iterations is 7 when there are 3 clients” seems completely out of place. This reviewer is not aware where the additivity property of gradients originates from. A reference would be very helpful. What is the intersection of gradients? How exactly using Eq. 8 removes the need for using all $2^n - 1$ coalitions?

6. Eq.s 7 and 8 also seem to assume that influence is additive across the data points from a single provider. That however is not the case, see [Koh et al](https://arxiv.org/abs/1905.13289).

7. The paper needs to spend a bit more time describing the datasets and why they are 1) appropriate and 2) sufficient for evaluating the proposed method. Are these generation tasks? Why were BLEU and ROUGE-* metrics used? Why not consider semantic similarity metrics?

**Questions:**

1. Please see weaknesses 1-6.
2. Given that gradients seem to be a gross approximation of the retraining procedure, how does this affect the quality of Shapley Values?
3. Section 3.1: Does one contributor always contribute data from one domain only?
4. Line 165: on the value of k and k’ not increasing: does it mean that after a provider introduces a duplicate dataset, the original as well as the duplicate would get the same score? Doesn’t it contradict line 79 which says that providers with highly similar data should be penalized?
5. Did the authors check that the evaluation datasets are not a part of the pretrained model training set?

---

> ### Author Response · Authors · 2024-11-25
> **Response to Reviewer Pz1e (Part 1/3)**
>
> Thanks very much for your time and effort in reviewing our paper and providing thoughtful feedback to improve our paper. We hope the following responses can clarify your confusion.
>
> **Q1: The technical novelty of the paper is somewhat limited. The Taylor expansion which underpins the gradient mechanism is already quite well-known in the influence function literature. The usage of Shapley values is also quite common in data valuation literature. The contribution of the paper seems to be a fast way to compute them but the paper does not provide sufficient detail on that computation (more on this point below).**
>
> **A1:**   We would like to defend the novelty of our framework since the purpose of our valuation methods is not to simply and blindly propose a new data selection or client contribution estimation approach.
>
> (1) Task Formulation: We are the first to propose a framework for measuring data value in the context of large models. We define both multi-source and single-source data settings and outline the task formulation and experimental scenarios for each setting.
>
> (2) Valuation Framework: From the perspective of data impact, we introduce a theoretical foundation based on gradient inner product, incorporating gradient projection and gradient correction. Additionally, in the multi-source data setting, we employ the Shapley-fast computation method. These components form a comprehensive framework for data valuation.
>
> (3) Empirical results: Through comprehensive experiments across different scenarios, we have validated the efficiency, accuracy, and robustness of our framework.
>
>
>
>
>
> **Q2: Eq. 5: Given that the loss is a mere proxy for the task accuracy, it is not clear how reliable the result of “summing up this formula in all the iterations in which was used to update the parameters” is. For instance, according to Guo et al, the accuracy and loss do not always go hand in hand. Does this difference between loss and accuracy, and the non-convexity of deep models have any effect on the reliability of the influence estimate? I think the paper would be greatly strengthened by adding some clarifications here.**
>
> **A2:**  Thanks. Loss is widely used as a direct optimization objective during training and serves as a common metric for evaluating model performance. In practical scenarios of instruction tuning, it remains the most frequently used indicator for model observation. Therefore, leveraging the impact of training data on the loss of validation data to assess the value of data is a reasonable approach. Directly using valuation data to train the model and then employing downstream metrics for valuation is costly and unstable.
>
> Although a lower loss does not necessarily guarantee better performance on downstream tasks, our experiments further provide empirical results. Similar to the setting in Table 2, the target domain is set to Consult and we select a 4000-sample subset from the original 15000-sample dataset with 3 different strategies: (a)-**Maximum Selection**, which selects samples with the largest data scores. (b)-**Minimum Selection**, which selects samples with the smallest data scores. (c)-**Random Selection**, which selects samples in a random manner. The filtered subset is then employed to fine-tune a ChatGLM3-6B model and test its downstream performance on target domains. The results, shown below, indicate that the data with the highest data score results in better downstream training performance. This also confirms the effectiveness of our method in active learning scenarios related to data selection.
>
> |                   | BLEU-4  | ROUGE-1 | ROUGE-2 | ROUGE-L |
> | ----------------- | ------- | ------- | ------- | ------- |
> | Maximum Selection | 13.2779 | 36.6102 | 12.9326 | 26.7862 |
> | Random Selection  | 13.0454 | 36.5104 | 12.6523 | 26.5085 |
> | Minimum Selection | 12.2375 | 35.7346 | 12.2955 | 25.9136 |
>
>
>
>
>
>
>
>
>
> **Q3: Eq. 6: The insight that the other parameters of the Adam optimizer should also be considered makes sense on a high level. However, the equation is discussed in a very standalone manner and leads to more questions than answers. Why the first two moments? What is meant by “calibration”? How does the equation generalize to related optimizers like AdamW and SGD+Momentum?**
>
> **A3:** In the theoretical derivation of Eq. (3), we assumed the use of Stochastic Gradient Descent (SGD) for updates. However, in practical applications of LLMs, adam-style optimizers are often employed for updates. Therefore, to enhance the theoretical self-containment, we need to adjust Eq. (3) to follow the adam update mechanism by maintaining first-order and second-order momentum estimates of the gradients. This adjustment ensures a more rigorous theoretical foundation. For other related optimizers, we only need to adjust Eq. (3) according to the specific update mechanism (e.g., maintaining additional momentum). In our approach, we employ a more general form of the adam optimizer for the calibration.

---

> ### Author Response · Authors · 2024-11-25
> **Response to Reviewer Pz1e (Part 2/3)**
>
> **Q4: The treatment of “domains” in the paper is a bit hand-wavy. Are domains so easy to define in the era of LLMs where the number of tasks one could perform with a LLM is virtually unlimited. Similarly, the update strategy seems to implicitly assume that we have similar number of data points from each domain (the way the losses for each data points are summed up over the iterations). Some domains, however, may contribute much less data and may have a higher loss. Does the paper suggest to train on these domains for longer? How does this impact the loss summation strategy?**
>
> **A4:** In our experimental setup, we primarily consider whether our framework can evaluate data from different sources across various target domains. To control variables, we collected data from different domains and ensured that the data volume from each domain was the same. We also conducted experiments in both coarse-grained and fine-grained scenarios.
>
> In the revision, we will include more realistic situations, such as varying data volumes for each domain, to better align with real-world scenarios.
>
>
>
> **Q5: The text below Eq. 7 is very difficult to parse. The phrase “the required number of training iterations is 7 when there are 3 clients” seems completely out of place. This reviewer is not aware where the additivity property of gradients originates from. A reference would be very helpful. What is the intersection of gradients? How exactly using Eq. 8 removes the need for using all coalitions?**
>
> **A5:** We use the contribution-weighted approach of marginal subsets to measure the value of data from different data providers. In traditional Shapley value methods, the model's performance metric after training is often used as the utility function, resulting in $(2^𝑛 − 1)$ marginal subsets for 𝑛 clients. In our optimized approach, we use the inner product of gradients (between the data to be evaluated and the support set) as the utility function. We find that such an approach can naturally leverage the additivity of gradients to improve efficiency. This additivity is based on the property that gradients can be summed when the model has not been updated (as discussed in detail in Q6 below).  As shown in Eq. (8), we can reuse computed gradients grad(Da) and grad(Db) from individual clients to calculate their union’s gradient, it is only necessary to recompute the overlapping portion.
>
>
>
>
>
> **Q6: Eq.s 7 and 8 also seem to assume that influence is additive across the data points from a single provider. That however is not the case, see Koh et al.**
>
> **A6:** (1) In our valuation framework, we will not perform model training or parameter updates; we will only compute the gradient of the entire sample set. Due to the additive mathematical property of gradient inner products, we can aggregate the gradients of all samples before applying gradient correction and projection. When batch_size=1, the sum of the gradients of different data points is equivalent to the total gradient of a batch of data (which is the same as the method used in training updates).
>
> (2) We carefully reviewed the reference provided by the reviewer. In section 2.1, it is mentioned that *"When measuring the change in test prediction or test loss, influence is additive. When measuring the change in self-loss, influence is not additive."* In our setup, we are estimating the influence of the evaluation data on the support set, which aligns with the definition of the change in test loss in the referenced document, and therefore is additive.
>
>
>
> **Q7: The paper needs to spend a bit more time describing the datasets and why they are 1) appropriate and 2) sufficient for evaluating the proposed method. Are these generation tasks? Why were BLEU and ROUGE-\* metrics used? Why not consider semantic similarity metrics?**
>
> **A7:** Thank you for your suggestions. We will include more details about the dataset in the revision, and we will also provide additional relevant details. For the evaluation section, BLEU and ROUGE are the most common metrics for LLM question-answering tasks, and we have strictly followed this protocol. In contrast, metrics like semantic textual similarity, which rely on embeddings for evaluation, often require extra storage for embeddings and are more commonly used to evaluate encoder models. We will include more results related to semantic similarity metrics in the revision.

---

> ### Author Response · Authors · 2024-11-25
> **Response to Reviewer Pz1e (Part 3/3)**
>
> **Q8: Given that gradients seem to be a gross approximation of the retraining procedure, how does this affect the quality of Shapley Values?**
>
> **A8**：Thanks. We adopt the gradient inner product as a data score for the utility function, and employ the standard Shapley value calculation method. In single-source scenarios, the calculation of the data score is decoupled from the Shapley value calculation in multi-source scenarios. Therefore, using the data score as an estimate of data impact **does not affect the quality** of the Shapley value. Specifically, it does not impact the properties of the Shapley value when evaluating data impact through marginal subsets. Our experiments also demonstrate that our valuation results are highly consistent with the ground-truth Shapley values.
>
>
>
> **Q9: Section 3.1: Does one contributor always contribute data from one domain only?**
>
> **A9:**  In our multi-source experiments, data from different providers comes from different domains, but each individual provider only possesses data from a single domain. We will add more scenarios in the revision, where each data provider has data from different domains. However, we believe that such scenarios do not fundamentally differ from the existing ones because, during the computation process, gradients are calculated for each sample individually and are additive.  We will add more discussions on this in our revision.
>
>
>
> **Q10: Line 165: on the value of k and k’ not increasing: does it mean that after a provider introduces a duplicate dataset, the original as well as the duplicate would get the same score? Doesn’t it contradict line 79 which says that providers with highly similar data should be penalized?**
>
> **A10:**  At line 165, the clone robustness we proposed refers to the scenario where, if a data provider has data valued at $w^k_0$, and a new data provider joins, duplicating the same data, then the value of the original and new data providers' data will be $w^k_1$ and $w^k_2$ respectively. In this case, we need to penalize such redundant clients dynamically, resulting in $w^k_1 = w^k_2 < w^k_0$. The value of the new client's data will significantly decrease after the penalty.
>
>
>
> **Q11: Did the authors check that the evaluation datasets are not a part of the pretrained model training set?**
>
> **A11:** We believe that this issue is a common challenge faced by all instruction tuning and reinforcement learning (RL) related work, as the companies that pre-train LLMs do not disclose the details of their pretraining data. However, we do not think this affects our valuation of the instruction tuning process. The data formats and loss calculation methods (e.g., masking) differ between pretraining and instruction tuning. The significant performance improvements observed when our data is used to train downstream LLMs indicate that these LLMs have a low fit to our data.

---

> ### Author Response · Authors · 2024-12-02
> **Sincerely Looking Forward to Your Feedback**
>
> Dear reviewer Pz1e, we are grateful for the valuable input you've provided. As we near the end of the discussion period, if you have any more questions or need clarification on our paper, please let us know. Thank you again for your time and consideration!

---

### Official Review · Reviewer_euoE · 2024-11-05

**Soundness:** 3
**Presentation:** 3
**Contribution:** 3
**Rating:** 5
**Confidence:** 3

**Summary:**

The paper introduces NESTLE, a data valuation framework. It combines gradient tracing with Shapley value concepts, enabling efficient, valuation across diverse domains and multiple data providers. Extensive experiments demonstrate that the proposed framework is capable of accurately and robustly providing accurate estimates of data value with a much lower cost over benchmarks.

**Strengths:**

- By using gradient tracing instead of traditional retraining, this method is more efficient and requires less computational power
- The framework’s low-cost gradient-based method scales well with model and dataset sizes, and it shows promising results across domains
- The method is capable of evaluating data from multiple providers

**Weaknesses:**

- The paper may benefit from rigorous writing, e.g., on page 2, the authors listed the design principles for a good data valuation framework, essentially a set of axioms, however, there’s no theoretical analysis to formally state that the proposed method satisfy these axioms. A theoretical justification would significantly enhance this paper

- The technical writing can be improved to enhance its readability.
    - Many notations are not defined, e.g., U and N on page 1 line 50, ηa_ bar in Eq (5), beta in eq (6)
    - References look odd, e.g., Kevin Fu Jiang, Weixin Liang, James Y. Zou, and Yongchan Kwon. Opendataval: a unified benchmark for data valuation. *In Alice Oh, Tristan Naumann, Amir Globerson, Kate Saenko, Moritz Hardt, and Sergey Levine (eds.),* Neurips, 2023a.

- Clearly state the computational requirement (see below) and contrast with existing work

**Questions:**

- Does this method provide fairness guarantees of Shapley values?
- Could you elaborate on the memory requirements for caching in relation to some common LLMs, as mentioned in line 218?
- In line 223, the calibration mechanism for the Adam-style optimizer requires the second-order moment. Could you describe the complexity and computational requirements involved?
- Based on Table 6, it appears that the absolute data value is less relevant than the relative value. For instance, even when 100% of the data is already in the training set, it retains a data value of 0.53. How could this be effectively applied in practice?
- To complicate matters, the non-comparable nature of value magnitudes across different LLMs means that some datasets may rank higher for certain models, making it more challenging to draw definitive conclusions. Any thoughts?

---

> ### Author Response · Authors · 2024-11-25
> **Response to Reviewer euoE (Part 1/2)**
>
> **Q1: The paper may benefit from rigorous writing, e.g., on page 2, the authors listed the design principles for a good data valuation framework, essentially a set of axioms, however, there’s no theoretical analysis to formally state that the proposed method satisfy these axioms. A theoretical justification would significantly enhance this paper**
>
> **A1:** Thanks for your suggestions! We will enhance our writing in the revision to ensure these sections are more comprehensible.
>
> (1) In our original submission,  we provided the design principles of the valuation framework in order to establish clear objectives for our method design. In our experiments, we have demonstrated its accuracy  (Tables 3, 4), efficiency (Table 5), and transferability (Tables 1, 2) of our valuation framework.
>
> (2) For the properties discussed in the robustness section, these properties are summarized from previous work, and the traditional Shapley value calculation method has already been proven to satisfy these properties (please see [1] for more details). In our valuation framework, we employ a Shapley-based computation in practical multi-source settings (utilizing the inner product of gradients as the utility function for the Shapley value). Such marginal-contribution-based computation method thus ensures that our approach maintains its robustness properties.
>
> We would like to thank the reviewer once again for their comments. We will improve our writing in the revision and include more theoretical justification.
>
> [1] Sim, Rachael Hwee Ling, Xinyi Xu, and Bryan Kian Hsiang Low. "Data Valuation in Machine Learning:" Ingredients", Strategies, and Open Challenges." IJCAI. 2022.
>
>
>
> **Q2: The technical writing can be improved to enhance its readability.**
>
> - **Many notations are not defined, e.g., U and N on page 1 line 50, ηa_ bar in Eq (5), beta in eq (6)**
> - **References look odd, e.g., Kevin Fu Jiang, Weixin Liang, James Y. Zou, and Yongchan Kwon. Opendataval: a unified benchmark for data valuation.** ***In Alice Oh, Tristan Naumann, Amir Globerson, Kate Saenko, Moritz Hardt, and Sergey Levine (eds.),*** **Neurips, 2023a.**
> - **Clearly state the computational requirement (see below) and contrast with existing work**
>
> **A2:**  We appreciate your comments. We will revise our notation to be more self-contained and add comparisons with recent related studies to enhance its readability.
>
>
>
> **Q3: Does this method provide fairness guarantees of Shapley values?**
>
> **A3:**   Thanks! For traditional standard Shapley values, the fairness is ensured through the use of marginal subsets, as proved in previous works [1].  In our framework, we strictly follow the standard Shapley value calculation and the only difference is the utility function. Specifically, we adopt the gradient inner product, i.e., data score, instead of the vanilla performance metric for the utility function. That is, in single-source scenarios, the calculation of the data score is decoupled from the Shapley-value-based adjustment in multi-source scenarios. Therefore, it does not affect the fairness of the Shapley value using the data score as the utility function to estimate data impact.
>
> [1] Jia, Ruoxi, et al. "Towards efficient data valuation based on the Shapley value." *The 22nd International Conference on Artificial Intelligence and Statistics*. PMLR, 2019.
>
>
>
> **Q4: Could you elaborate on the memory requirements for caching in relation to some common LLMs, as mentioned in line 218?**
>
> **A4:**  Thanks for your question. Considering an LLM with (k)-Billion of parameters, and using LoRA for fine-tuning where the proportion of trainable parameters is $t%$, the shape of each original gradient to be cached (whether it's per-sample or the total gradient for a client batch) is  $(\frac{t}{100}) \times k \times 10^9$. Therefore, a tensor of this shape would occupy around  $({t} \times k \times 40)$ MB of memory with 32-bit precision (Typically, when finetuning a 7B model with LoRA ranking=8, then $k=7$, $t\approx0.3$ and the memory is around 96MB). Further, In our framework, we further adopt a projection strategy to reduce the original gradient dimensions to only 4096, thus the memory requirements for such a [1,4096] tensor is reduced to merely 16 KB with 32-bit precision for all LLMs.

---

> ### Author Response · Authors · 2024-11-25
> **Response to Reviewer euoE (Part 2/2)**
>
> **Q5: In line 223, the calibration mechanism for the Adam-style optimizer requires the second-order moment. Could you describe the complexity and computational requirements involved?**
>
> **A5: Actually,** such calibration mechanism has negligible impact on the overall computational and storage requirements. In our original SGD assumption of Eq.(3) and Eq.(5), we only need to compute the gradient for each to-be-valuated sample/batch and apply a fixed learning rate directly. When using the Adam-style calibration, besides the gradient computation for each sample/batch, We need to additionally compute the squared term of the corresponding gradient (this only requires one multiplication operation). In addition, we require two extra tensor variables to store the momentum of the first-order and second-order gradients, similar to the Adam optimizer. As mentioned above in Q4 above, each of these two global moving average gradient momenta would merely occupy around $({t} \times k \times 40)$ MB of memory with 32-bit precision (These two momenta are global and shared by all samples.). Hence, this calibration process does not introduce too much additional computational complexity.
>
>
>
>
>
> **Q6: Based on Table 6, it appears that the absolute data value is less relevant than the relative value. For instance, even when 100% of the data is already in the training set, it retains a data value of 0.53. How could this be effectively applied in practice?**
>
> **A6:** The primary purpose of conducting the experiments presented in Table 6 is to prove that our valuation framework assigns lower data scores to data that the LLM has already been trained on, which corresponds with real-world observations.
>
> In practical situations,  it does not impact real-world usage data that already-trained data still maintains a data score above zero. We can additionally set a threshold to differentiate between samples that provide an actual benefit to an LLM and those that do not. Specifically, the data score of data already fitted by the LLM can serve as a reference for setting a threshold, and we can directly use the data score of such trained samples as the threshold for distinguishing between beneficial and non-beneficial samples.
>
>
>
> **Q7: To complicate matters, the non-comparable nature of value magnitudes across different LLMs means that some datasets may rank higher for certain models, making it more challenging to draw definitive conclusions. Any thoughts?**
>
> **A7:** Thank you for your insightful question. The fundamental goal of our valuation framework is to estimate the data value for a particular LLM in different target domains. Due to the different existing capabilities of various LLMs in different domains, the value rankings for datasets will naturally vary (e.g., when model A has superior performance in domain D relative to model B, then the data from domain D might be less critical for model A than for model B). Hence such phenomenon is normal and reasonable that the value rankings of different datasets vary across different LLMs.
>
> Further, if we want to decouple the impact of LLM and consider the data value across various target domains (rather than for a specific LLM), a simple approach would be to consider multiple representative, un-finetuned LLMs and average their scores through a voting system. We will add more analysis and discussion on this topic in our revision.

---

> ### Author Response · Authors · 2024-12-02
> **Sincerely Looking Forward to Your Feedback**
>
> Dear reviewer euoE, thanks again for providing valuable suggestions! As we are close to the end of the discussion period, in case you have further questions about our paper, please don't hesitate to let us know. Thank you again for your time and consideration!

---

### Meta-Review · Area_Chair_eueM · 2024-12-16

**Metareview:**

The paper presents an approach to estimate the value of data sources for language model training. The approach is based on a Shapley value estimate, which in turns relies on a form of gradient tracing as model training progresses. The paper details how to efficiently handle multiple data sources in this framework.

Reviewers raised a number of concerns on the work, primarily around the limited novelty compared to prior methods such as LESS (and more broadly those based on Shapley values), and lack of clarity in technical presentation and writing. Following the response, there was still some confusion on whether the distinction to LESS is significant enough. From the AC's reading, this concern appears justified; while the starting point is motivated from influence functions, the final proposal in Equation 5 is indeed highly similar to that of LESS and related methods such as TracIn. It is not apparent that the discussions in Section 3.3 suffice as a standalone contribution of technical depth. Given this, we believe it is challenging to accept the paper.

**Additional Comments On Reviewer Discussion:**

The initial reviews raised a large number of concerns, most notably:

(1) the similarity of the proposal to prior works such as LESS

(2) the lack of clarity in the technical presentation and writing

For (1), the author response argued that the proposal involves an application to a new task (domain selection versus data selection), and requires a less onerous training process. The first argument is valid, but does not assuage concerns about the technical novelty; further, a reframing of the claimed contributions may be appropriate in the abstract and introduction. The second argument also may be valid, but again connects to the first in that the precise application to data selection involved some warm-up in LESS, while the core idea of estimating value based on gradient tracing remains.

For (2), the author response made some clarifications. These would be useful to incorporate in the text of an updated version of the paper.

Overall, following (1), the concerns about technical novelty remained. From the AC's reading, it is indeed not fully clear how technically new the paper's contributions are.

---

### Decision · Program_Chairs · 2025-01-22

Reject